# Paleoclimate data assimilation with CLIMBER-X: An ensemble Kalman filter for the last deglaciation

**Ahmadreza Masoum**[1,2]*, **Lars Nerger**[1], **Matteo Willeit**[3], **Andrey Ganopolski**[3], **Gerrit Lohmann**[1,2]

1 Section Paleoclimate Dynamics, Alfred Wegener Institute Helmholtz Centre for Polar and Marine Research, Bremerhaven, Germany, 2 Center for Marine Environmental Sciences, University of Bremen, Bremen, Germany, 3 Department of Earth System Analysis, Potsdam Institute for Climate Impact Research, Potsdam, Germany

* ahmadreza.masoum@awi.de

## Abstract

Using the climate model CLIMBER-X, we present an efficient method for assimilating the temporal evolution of surface temperatures for the last deglaciation covering the period 22000 to 6500 years before the present. The data assimilation methodology combines the data and the underlying dynamical principles governing the climate system to provide a state estimate of the system, which is better than that which could be obtained using just the data or the model alone. In applying an ensemble Kalman filter approach, we make use of the advances in the parallel data assimilation framework (PDAF), which provides parallel data assimilation functionality with a relatively small increase in computation time. We find that the data assimilation solution depends strongly on the background evolution of the decaying ice sheets rather than the assimilated temperatures. Two different ice sheet reconstructions result in a different deglacial meltwater history, affecting the large-scale ocean circulation and, consequently, the surface temperature. We find that the influence of data assimilation is more pronounced on regional scales than on the global mean. In particular, data assimilation has a stronger effect during millennial warming and cooling phases, such as the Bølling-Allerød and Younger Dryas, especially at high latitudes with heterogeneous temperature patterns. Our approach is a step toward a comprehensive paleo-reanalysis on multi-millennial time scales, including incorporating available paleoclimate data and accounting for their uncertainties in representing regional climates.

## Introduction

During the last deglaciation 20–10 kyr BP (kyr = 1000 years; BP = before present), the climate warmed due to solar insolation, melting ice sheets, and increasing concentrations of carbon dioxide ($CO_2$) as well as other greenhouse gases ([1]). The last deglacial is a key phase for understanding abrupt climate shifts associated with changes in ocean circulation at the end of the last ice age ([1–8]). Despite several proxy-based reconstructions of surface temperature

**Funding:** AM and GL receive funding through Center for Marine Environmental Sciences at the University of Bremen (https://www.marum.de/en/Research/MARUM-Cluster-of-Excellence-Projects-OC.html), "Ocean and Cryosphere under climate change" in the Program "Changing Earth Sustaining our Future" of the Helmholtz Society (https://www.helmholtz.de/en/about-us/structure-and-governance/program-oriented-funding/) and through PalMod by the Bundesministerium für Bildung und Forschung (https://www.palmod.de/; grant no. 01LP1917A). The funders had no role in study design, data collection and analysis, decision to publish, or preparation of the manuscript.

**Competing interests:** The authors have declared that no competing interests exist.

being available (e.g. [1, 9]), a coherent pattern is still unknown, and climate model simulations (e.g. [4, 8, 10–13]) also show inconsistent results.

An alternative method to the above-mentioned methods for reconstructing past climates is data assimilation (DA), which merges the information from models and proxy-based reconstructions. DA has been recently applied to reconstruct, for instance, the climate of the past millennium (e.g. [14–21]), last glacial maximum (LGM; e.g. [22, 23]), Younger Dryas (YD; [24]), and for the last glacial termination ([25]). The motivation behind DA is to jointly use model simulations and paleoclimate information to estimate the most likely state and trajectory. In a DA system, the data provide climate information at the sites, and the model fills in the missing information about the other locations by adding model dynamics and error bars from the data and the model ([26]).

Previous studies of DA in paleoclimate have focused primarily on the Holocene (e.g. [21, 27]). Moreover, they largely focused on reconstructing one observed variable, mainly surface temperature, and did not investigate the effect of DA on the model performance in simulating other climate variables. In general, they employed the offline approach, in which the initial conditions for the next DA-cycles are not analyzed (e.g. [28]) and used different existing simulations as the model background states (e.g. [14, 25]) in their DA system to avoid the enormous computational cost of long-term transient simulations. However, they did not determine how the quality of the model states, which can be influenced by boundary settings, affects DA solutions for past climate reconstructions. [14] explore the significance of selecting a time-varying or time-constant model background in the offline DA.

In this study, we assimilate surface temperatures (ST) from oceanic and terrestrial paleoclimate records of the recent deglaciation using the newly developed Earth system model of intermediate complexity (EMIC), CLIMBER-X ([29]). EMICs are simplified tools for studying the long-term dynamics of the Earth system ([30]) that do not explicitly resolve internal variability on synoptic to centennial time scales but are suitable for long-term integrations of the climate system. CLIMBER-X is capable of simulating ≈10,000 years per day, making it suitable for transient runs for the last termination. Here, the model is also equipped with a Parallel Data Assimilation Framework (PDAF; [31]), providing an efficient variant of the ensemble Kalman filter algorithm ([32]) as well as a stochastic emulator mimicking internal variability in the model. Our primary objective in conducting our modelling exercises is to assess the efficacy of our DA technique in CLIMBER-X. Moreover, to evaluate the uncertainty of different ice sheet reconstructions, we conduct our experiments with respect to two available ice sheet reconstructions and their deglacial meltwater entering the ocean. This uncertainty is motivated by the predominant influence of ice sheet reconstructions on the timing and occurrence of climate events ([13, 33]). Finally, we evaluate the effects of DA on climate trajectories and compare the results for the different ice sheet reconstructions to understand the influence of background states in the DA system.

This paper is organized as follows. The next section briefly overviews our climate model, the DA algorithm, and the experimental design. We present and discuss the results in the sections thereafter. Finally, our conclusions are drawn in the last section.

## Methods

### Model

CLIMBER-X is an Earth system model of intermediate complexity (EMIC; [30]). CLIMBER-X has the horizontal resolution of 5˚ × 5˚ and employs the semi-empirical statistical–dynamical atmosphere model (SESAM; [29]), the 3-D frictional–geostrophic ocean model GOLDSTEIN ([34–36]), the thermodynamic sea ice model (SISIM; [29]), and the land surface model

PALADYN ([37]) as the submodels to simulate different climate components. CLIMBER-X is designed to simulate the mean climatological state and is almost 1000 times faster than full General circulation models ([29]).

## PDAF

PDAF provides a computationally efficient framework for performing ensemble-based DA with different filters in numerical models (http://pdaf.awi.de; [31]). It separates the data assimilation system into three parts: numerical model, observations, and filter algorithms. The filter algorithms combine the model and observational information.

PDAF can be efficiently coupled with numerical models allowing us to have a model with data assimilation extension. The high efficiency and full parallelization features of PDAF make it a proper tool for different ensemble sizes for data assimilation in climate models ([38]). Different types of ensemble-based Kalman filter algorithms, particle filters and variational methods for DA are available in PDAF. We choose the Local Error Subspace Transform Kalman Filter (LESTKF) as the DA algorithm in this study because it is a particularly efficient formulation for high-dimensional DA ([39]).

The LESTKF is a localized version of the Error Subspace Transform Kalman Filter (ESTKF; [39]). The ESTKF uses an ensemble of $m$ model states of size $n$, which are stored as columns of the matrix $\mathbf{X}_k$. The prior climate states matrix $\mathbf{X}_k^{prior}$ from the model simulations is converted into a matrix of analysis states $\mathbf{X}_k^a$ at time $t_k$ using the transformation

$$\mathbf{X}_k^a = \bar{\mathbf{x}}_k^{prior}\mathbf{1}_m^T + \mathbf{X}_k^{prior}(\mathbf{w}_k\mathbf{1}_m^T + \tilde{\mathbf{W}}_k). \tag{1}$$

Here, $\bar{\mathbf{x}}_k^{prior}$ presents the prior ensemble mean state of size $n$, and $\mathbf{1}_m$ is a vector of size $m$ having the value of one in all elements. Additionally, $\mathbf{w}_k$ is a vector size of $m$ transforming the ensemble mean, and the ensemble perturbation is transformed by the matrix $\tilde{\mathbf{W}}_k$ of size $m \times m$ named weight matrix. Since all computations in the analysis refer to the time $t_k$, the time index $k$ is skipped hereafter.

The ensemble transformation matrix and vector are calculated in an error subspace of dimension $m - 1$ represented by the prior ensemble. An error-subspace matrix can be computed by $\mathbf{L} = \mathbf{X}^{prior}\mathbf{T}$, where the matrix $\mathbf{T}$, named projection matrix, has the size of $m \times m - 1$ and is defined by

$$\mathbf{T}_{ji} = \begin{cases} 1 - \dfrac{1}{m}\dfrac{1}{\frac{1}{\sqrt{m}} + 1} & \text{for } i = j, j < m \\[2em] -\dfrac{1}{m}\dfrac{1}{\frac{1}{\sqrt{m}} + 1} & \text{for } i \neq j, j < m \\[2em] -\dfrac{1}{\sqrt{m}} & \text{for } j = m. \end{cases} \tag{2}$$

The relation between the prior state vector $\mathbf{x}^{prior}$ and the vector of observations $\mathbf{y}$ is described as

$$\mathbf{y} = \mathbf{H}(\mathbf{x}^{prior}) + \epsilon, \tag{3}$$

where $\mathbf{H}$ is the observation operator, and $\epsilon$ gives the vector of observation errors, which is considered a white Gaussian distributed random process with observation error covariance matrix

**R**. For the analysis step, a transform matrix, which has size $(m-1) \times (m-1)$, is calculated as

$$\mathbf{A}^{-1} = \rho(m-1)\mathbf{I} + (\mathbf{HX}^{prior}\mathbf{T})^T\mathbf{R}^{-1}\mathbf{HX}^{prior}\mathbf{T}. \tag{4}$$

Here, the $\mathbf{I}$ is the identity matrix, and $\rho$ is named the "forgetting factor" ([40]). $\rho$ with the value of $0 < \rho \le 1$ is used to inflate the prior error covariance matrix. The weight vector $\mathbf{w}$ and matrix $\tilde{\mathbf{W}}$ are now defined by

$$\mathbf{w} = \mathbf{TA}(\mathbf{HX}^{prior}\mathbf{T})^T\mathbf{R}^{-1}(\mathbf{y} - \mathbf{H\bar{x}}^{prior}), \tag{5}$$

$$\tilde{\mathbf{W}} = \sqrt{m-1}\,\mathbf{TA}^{1/2}\mathbf{T}^T, \tag{6}$$

where $\mathbf{A}^{1/2}$ is the symmetric square root of $\mathbf{A} = \mathbf{US}^{-1}\mathbf{U}^T$ that is calculated from the eigenvalue decomposition $\mathbf{USV} = \mathbf{A}^{-1}$ such that $\mathbf{A}^{1/2} = \mathbf{US}^{-1/2}\mathbf{U}^T$.

For localization, which is required for a high-dimensional model, each individual grid point is independently updated by a local analysis step that considers only observations within a horizontal radius of influence $l$. Thus, a local observation operator computes an observation vector within the radius $l$ from the global model state. Furthermore, each observation is weighted according to its distance from the grid point. The weight is applied by multiplying the entries of matrix $\mathbf{R}^{-1}$ in the Eqs (4) and (5) by a weight which decreases from one to zero with increasing distance. The localization weight is computed by a fifth-order polynomial with a form similar to a Gaussian function ([41]). In localization, Eq (1) is used with individual matrices $\mathbf{w}_k$ and $\tilde{\mathbf{W}}_k$ for each local analysis region.

## Observations

We use the dataset provided by [9] as observations in the data assimilation system. The dataset includes well-dated temperature records from the last glacial period, including sixty-seven records from the ocean, interpreted as sea STs, and thirteen records for temperatures at the land surface. The dataset contains the absolute temperature values of each proxy site (green dots in Fig 2), the published age, and the corresponding errors of the age model ($\sigma$).

For the reconstruction of the early Holocene global mean surface temperature (GMST) anomaly (11.5–6.5 ka BP; ΔGMST), [9] first project the dataset onto a 5˚ × 5˚ grid, then linearly interpolate it to 100-year resolution and integrate it as area-weighted averages. The details of the age control, proxy temperatures, and uncertainty analysis are explained in [9]. As the spatial resolution of this reconstruction corresponds to that of CLIMBER-X, a comparison of the simulated trajectory with Shakun et al.'s reconstruction can be made with ease.

Since the dataset does not contain ST errors for the proxy sites, we translate the age model uncertainties into temperature uncertainties. To obtain the temperature uncertainty at time $t$ for each record, we subtract the corresponding temperature at $t - \sigma$ and $t + \sigma$ from the temperature of $t$ and take the absolute average of these variances as the temperature uncertainty in $t$. Finally, the average of the temperature uncertainties over all records was taken to represent the vector of observational errors in our DA system for every 100 years.

## Experimental design

The combination of our model with PDAF can simultaneously run the transient simulation and DA without restarting the model. Two experiments are performed. Exp_GLAC1D uses GLAC-1D ([42]) for the ice sheet reconstruction, bathymetry, and land-sea mask while a new ice sheet reconstruction, PaleoMist ([43]), is employed in Exp_PaleoMist. In both experiments, greenhouse gases and orbital forcing are taken from [44, 45], respectively. The GHG forcing

(S3 Fig) and the ice sheet reconstructions (S1 and S2 Figs) are shown in the supporting information.

The transient simulations start at 25 kyr BP with pre-industrial equilibrium and then switch to LGM boundary conditions. The model is subsequently run until the year 6.5 kyr BP using prescribed time-varying topography, bathymetry, ocean, greenhouse gases (GHGs), and orbital forcing. Adequate equilibrium and representation of the climate states at 22 kyr BP are achieved after three thousand years.

In our setup, data are assimilated into the model every 100 years from 22 to 6.5 kyr BP, which means that the DA system consists of 156 cycles. An implicit assumption in a DA system for the optimal combination of model predictions and observations is that data and model errors are random with a mean of zero (i.e. unbiased), indicating the importance of identifying and correcting observational errors before implementing DA ([46, 47]). In order to avoid systematic errors, the state vector containing the field updated by DA is initialized by yearly-averaged ST anomalies from the early Holocene (ΔST) at the last year of 100-year intervals in which the observation information is available (Fig 1). We take the values of early Holocene from the free run and use the prognostic variable, skin temperature, for calculating yearly-averaged ST. Another advantage of assimilating anomalies is that we can easily compare our DA results with the GMST reconstruction of [9], which is presented as an anomaly from the early Holocene.

In our DA system, the observation operator **H** in Eq (3) is a simple transformation matrix because the observation and the state vector have the same unit. This approach in paleoclimate DA is known as indirect DA ([48]). Indeed, **H** extracts ΔST at the observed states and subtracts the mean to obtain the analysis states. Moreover, our DA system is online ([49]), estimating the time-averaged state and initial condition for subsequent DA cycles (Fig 1). Accordingly, we compute an increment term defined as $\mathbf{X}_k^a - \mathbf{X}_k^{prior}$ and add it to the model field to calculate the state of next time step. In other words, the model has an updated (or corrected) initial field for integration through the next 100 years until the next update for the initial condition.

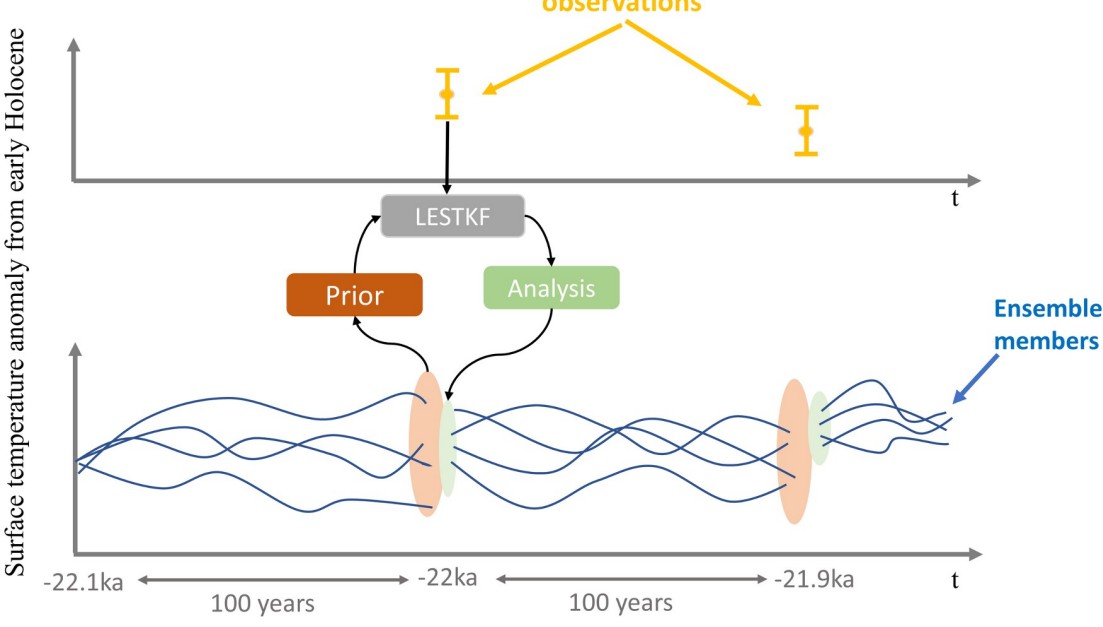

**Fig 1. Online DA.** Schematic view of our DA system for the first two cycles.

Many studies demonstrate that the predictability of surface temperatures, typically captured by most proxies, extends beyond an annual timeframe (e.g., [50, 51]). Consequently, in principle, online DA is expected to outperform offline DA when the model exhibits predictability that surpasses the averaging time represented by observations. This advantage arises from the ability of online DA, particularly in EnKF-based methods, to utilize more accurate initial conditions. To effectively leverage information from initial conditions with online DA, models must incorporate slowly changing components, such as the ocean model [49]. [52] suggest when the computational cost of online and offline methods is comparable, the online approach is generally preferable due to the temporally consistent states it offers. Given that CLIMBER-X incorporates an ocean component and is characterized by its efficiency as a fast model, we opt for the online DA approach. Furthermore, the online DA provides the added benefit of enabling us to evaluate the performance of CLIMBER-X in simulating variables beyond surface temperature.

In our approach, the DA system has 16 ensemble members, and the localization radius is 5000 km. The size of the ensemble was chosen in view of the computational cost. This choice is also consistent with [53], who have shown that an ensemble size of 15 or more is sufficient to constrain the simulations with the available proxy information.

Further, we determine the optimal localization radius by comparing experiments with different radii. The localization radius is an important factor, usually tuned individually for each application in the DA methods applying localization. The previous studies based on the observation network, prior states, their DA methods, and the goal of their reconstruction define their criteria for the optimal radius. For example, [49] employs an EMIC for their online DA experiments and uses a localization radius ranging from 2000 km to 8000 km. However, some other studies, which conduct the offline DA, select a relatively large radius localization such as 12000 and 25000 km (e.g., [14, 25, 23, 54]). The key factor guiding our choice of radius is its impact on the ST field. In Fig 2, we compare the DA result of the ST field after the first cycle of

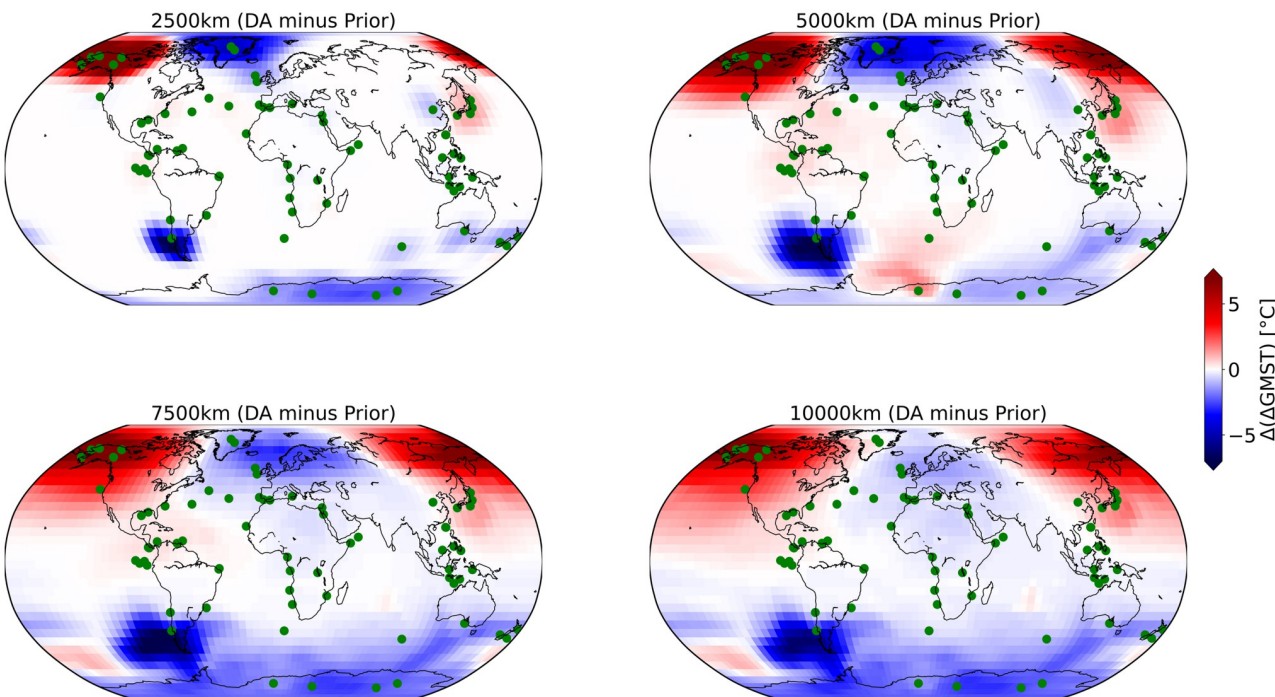

**Fig 2. Effect of DA using different radius.** Effect of DA increment at first analysis step on ST field using 2500, 5000, 7500, and 10000 km radius.

four DA experiments with radii of 2500, 5000, 7500, and 10000 km. The data assimilation effect appears to be too localized for a radius of 2500 km. Using a radius of 5000 km, almost all grid points (99%) are influenced by at least one observation while avoiding large-distance covariances beyond this radius. For the 7500 and 10000 km radii, the DA combined information from locations that are too distant, which leads to reduced effect, e.g. over Greenland, but also spurious effects like the rather uniform cooling of the Antarctic.

### Stochastic model component

In an ensemble-based DA system, the ensemble members represent the model uncertainty ([32]). Since CLIMBER-X is a deterministic model ([55]), we integrate ensemble members in parallel, each ensemble member on a single compute node with $2 \times 18$-core CPUs on a high-performance computer (Cray CS400 Xeon E5–2697V4 3.60 GHz), and add random perturbations to these model states at each model time step to obtain sufficient ensemble dispersion. We use ST values from a climate run in the TraCE-21000 project ([10]) to generate spatially-correlated perturbations as follows. First, the dataset is linearly interpolated to the spatial resolution of CLIMBER-X. Then snapshots of the ST values, from 22 to 6.5 kyr BP every 100 years are collected in a matrix $\mathbf{Z}$ with 156 columns, each containing the anomaly values in the grid points at a given time. After subtracting the temporal mean, we obtain the matrix $\mathbf{Z}'$. Then, the singular value decomposition $\mathbf{Z}' = \mathbf{VSW}$ is computed, yielding 155 empirical orthogonal functions (EOFs) stored in the columns of $\mathbf{V}$, while $\mathbf{S}$ is a diagonal matrix holding the corresponding singular values. Then, following second-order exact sampling ([56]), we compute the matrix of ensemble perturbations as

$$\Delta\mathbf{X} = \sqrt{m-1}\,\mathbf{SV}\boldsymbol{\Omega}^T. \tag{7}$$

Here $m$ is the ensemble size, and $\Omega$ is a random matrix that preserves the mean and the covariances. Consequently, the ensemble perturbations are added to the ensemble members following the autoregressive method ([57]) to make the model stochastic, providing an ensemble scatter. We add the perturbations to a prognostic variable named near-surface atmosphere temperature ($tam$) as

$$tam_k = tam_{k-1} + \varepsilon_k, \tag{8}$$

where $\varepsilon_k$ is perturbation at time $k$ defined as $(1 - \alpha)\Delta x_{k-1} + \alpha\Delta x_k$. $\Delta x$, containing perturbations for the grid points, is one column of matrix $\Delta\mathbf{X}$. Therefore, $\Delta\mathbf{X}$ has 16 different columns that are used by ensemble members. $\alpha$ is a user-defined coefficient, which is equal to 0.5 in our experiments. For the prior step, Fig 3 shows the ensemble spread for $\Delta$GMST. This perturbation method yields that the standard deviation of the ensemble spread for both experiments varies between $\approx 0.2°$ and $0.4°C$ during the experiments.

## Results

We start by comparing the outcomes of free model runs based on the GLAC1D and PaleoMist ice sheet reconstructions without DA. Thereafter, we analyze the effect of DA on $\Delta$GMST trajectories. Lastly, we show how the DA solution can alter the spatial patterns of the ST fields.

### Free runs

Fig 4a shows the absolute values of GMST for free runs with GLAC-1D and PaleoMist. The different time intervals LGM, Oldest Dryas (OD), Bølling-Allerød (BA), YD, and early Holocene are used here as in [9]. Both trajectories are nearly identical during the LGM. However,

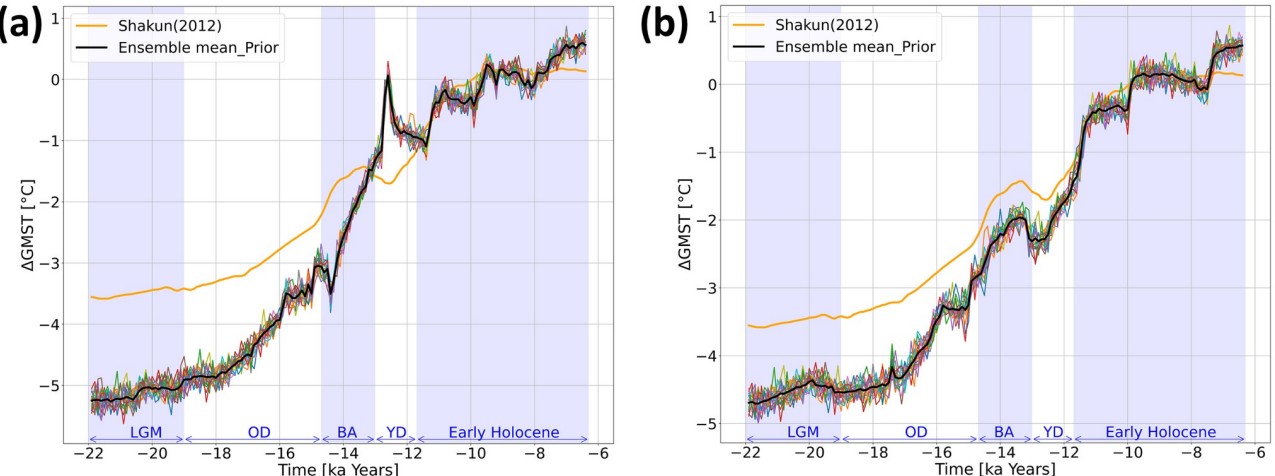

**Fig 3. Ensemble members before DA.** Ensemble members (coloured lines) and ensemble mean before DA in (a) Exp_GLAC1D and (b) Exp_PaleoMist.

during the rest of the time, the PaleoMist simulation shows a cooler GMST, except for the beginning of the BA interval. Moreover, the GMST in the PaleoMist simulation increases more steadily, while in the GLAC-1D simulation, there are two abrupt shifts in the GMST at the beginning and end of the BA. The differences in the magnitude, timing, and speed of the warming and cooling trends during BA and YD are evident between the two free simulations, suggesting that ice sheet reconstruction significantly impacts the transient simulation.

Comparing the transient simulations with the GMST reconstruction of [9], which has been projected to a 5˚ × 5˚ grid (Fig 4b), the ΔGMSTs simulated by the model are about 1.3–1.5 times colder than the proxy-based reconstruction during the LGM. [9] and Exp_PaleoMist_Free show a continuous warming trend during BA, while this trend is disrupted in Exp_GLAC-1D by a strong cooling followed by remarkably rapid warming. The trajectories of [9] and

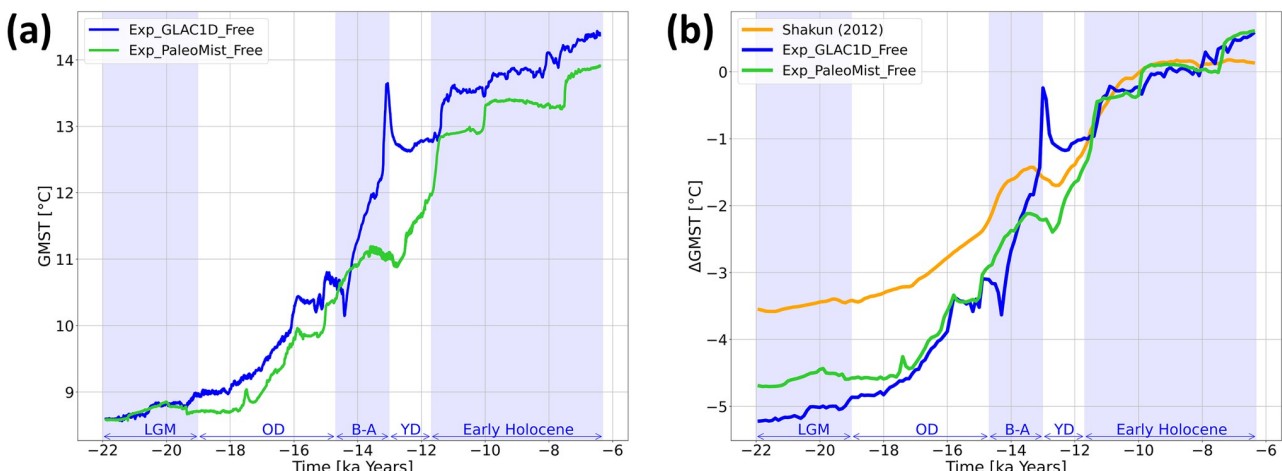

**Fig 4. GMST from free runs of the last deglaciation and comparison with the proxy-based reconstruction.** a) GMST from free runs of the last deglaciation using the different ice sheet reconstructions, GLAC-1D and PaleoMist. b) Comparison of GMST anomaly from the early Holocene of our free runs compared to [9].

Exp_PaleoMist_Free behave similarly over the YD. In contrast, Exp_GLAC-1D shows a strong decrease in ΔGMST beginning with the onset of the YD.

## Trajectories after DA

After applying the DA, we evaluate the ensemble members to ensure the DA system functions. The standard deviation of the ensemble is reduced between ≈ 20% and 70% in both experiments during DA cycles, showing that the DA system efficiently influences the ensemble members. Moreover, we compute the mean surface temperature anomaly from the early Holocene (ΔMST) before and after applying DA by averaging over the proxy sites and comparing it to the proxy-based ΔMST (Fig 5). The DA results are mainly between the observations and the simulated ΔMST. In Exp_GLAC1D (Fig 5a), the intensity of the abrupt changes in BA and YD declines, and ΔMST aligns more closely with the observations until the end of YD. As with Exp_PaleoMist, the DA trajectory exhibits slightly higher temperatures than both observation and the ensemble mean of the prior during LGM. However, the trajectories closely follow the same pattern for BA and YD in Exp_PaleoMist.

To further analyze the deglacial dynamics, Figs 6 and 7 display the net North Atlantic surface freshwater flux (FW), global sea surface salinity (SSS), Atlantic meridional overturning circulation (AMOC) at 26°N, and ΔGMSTs for Exp_GLAC1D and Exp_PaleoMist, respectively. Furthermore, we compare the ensemble mean of DA and prior states with the free run and a DA-based reconstruction conducted by [25] (Figs 6d and 7d). Comparing the prior ensemble mean and the free run in Exp_GLAC1D (Fig 6d), the abrupt warming shift in the onset of YD reaches its maximum with a 300 years delay. However, the prior ensemble mean and free run trajectories are similar in Exp_PaleoMist, except for a slightly warmer prior mean state during BA (Fig 7d).

In both experiments, the average SSS during the LGM is about one psu higher than during the early Holocene (Figs 6b and 7b), which is due to freshwater added to the ocean by melting ice sheets ([1, 58]). An increase in FW leads to a decrease in AMOC strength. For example, the sudden increase in FW at the beginning of BA in Exp_GLAC1D leads to an off-state in AMOC with a rapid increase at the end of the BA (Fig 6a and 6c) which is more difficult to reconcile with proxy data [59].

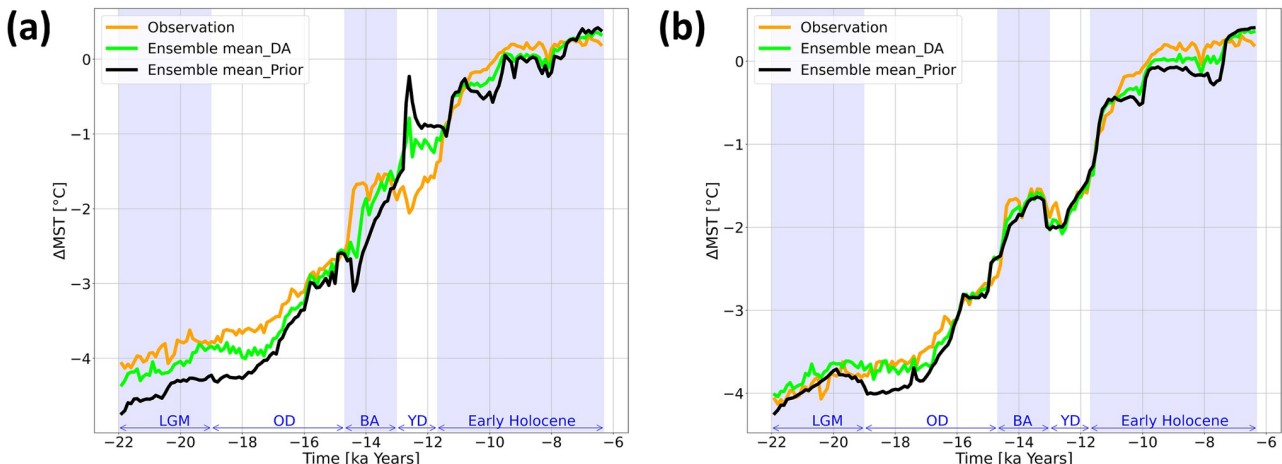

**Fig 5. ΔGMST calculated by raw averaging over the observation locations.** Mean surface temperature change (ΔMST) calculated by averaging over the proxy locations for (a) Exp_GLAC1D and (b) Exp_PaleoMist. The orange, green, and black lines illustrate trajectories for observation, DA ensemble, and prior ensemble means, respectively.

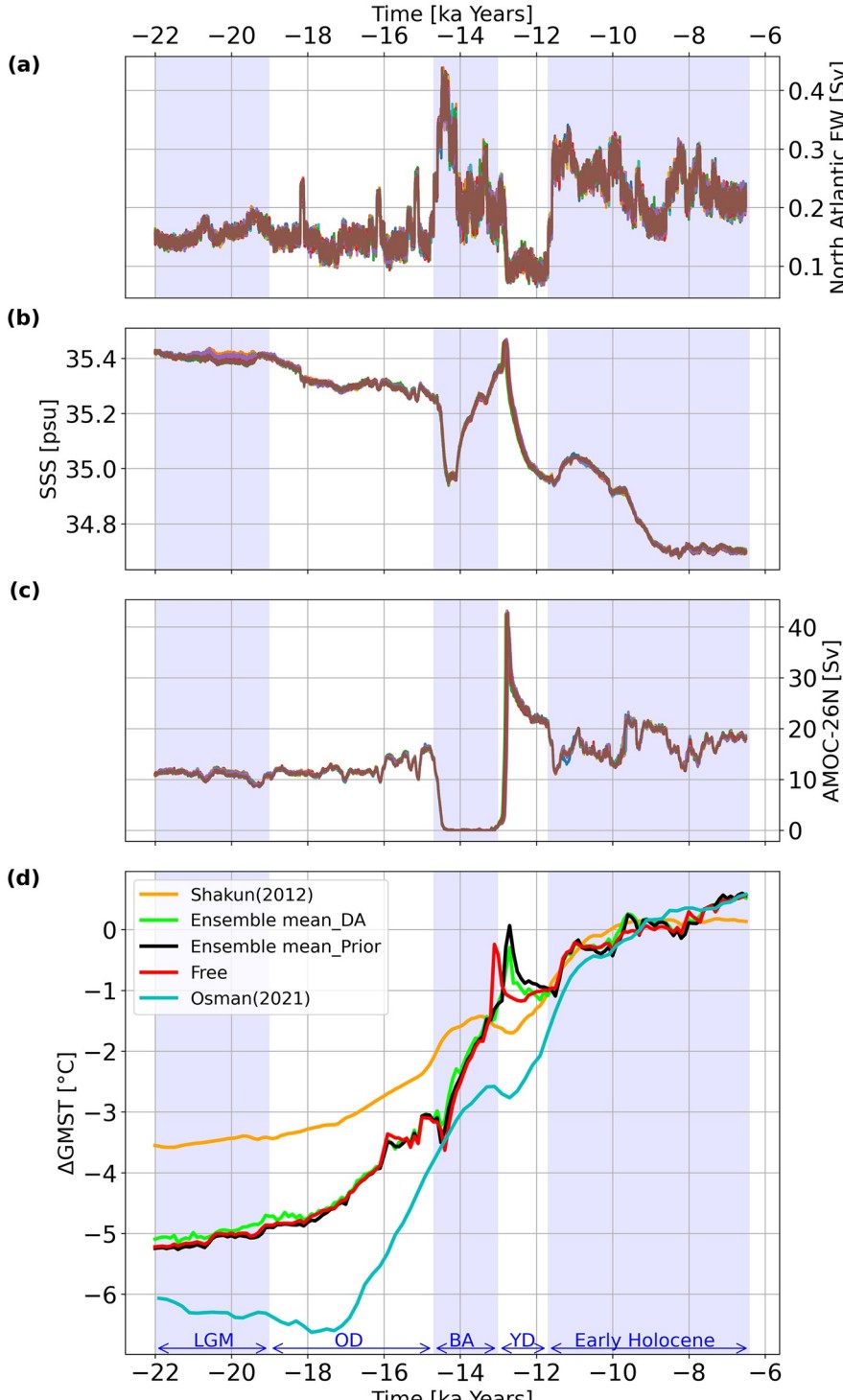

**Fig 6. Climate variables trajectories in Exp_GLAC1D.** North Atlantic FW (a), SSS (b), AMOC at 26° north (c), and ΔGMSTs based on the model field (d) for Exp_GLAC1D. Different colours in (a), (b), and (c) correspond to ensemble members. The red line in (d) represents ΔGMST for the free run. The orange line in (d) is the proxy reconstruction of ΔGMST by [9] projected to 5° × 5° resolution. The blue line in (d) is the DA-based reconstruction of ΔGMST by [25].

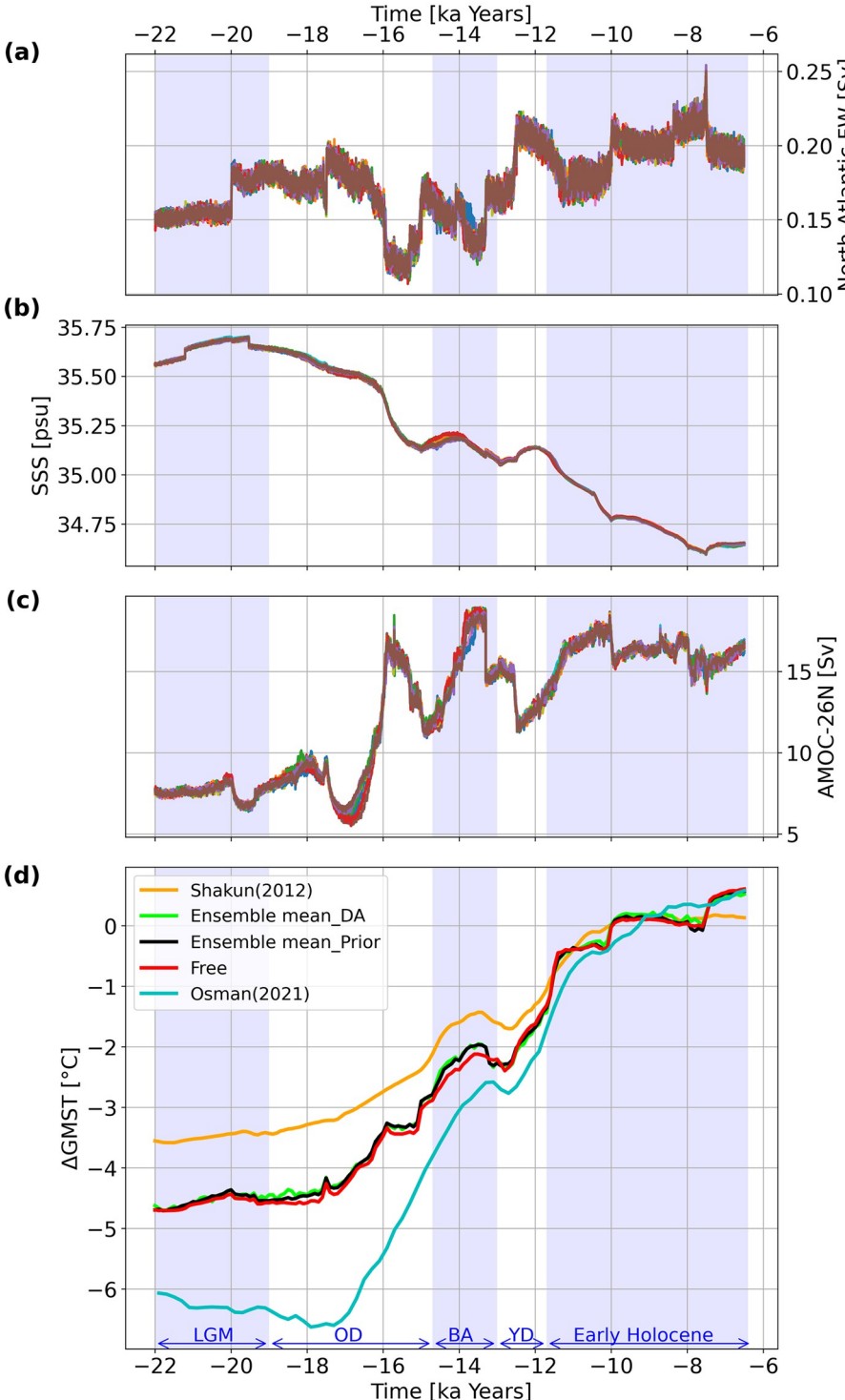

**Fig 7. Climate variables trajectories in Exp_PaleoMist.** North Atlantic FW (a), SSS (b), AMOC at 26˚ north (c), and ΔGMSTs based on the model field (d) for Exp_PaleoMist. Different colours in (a), (b), and (c) correspond to ensemble members. The red line in (d) represents ΔGMST for the free run. The orange line in (d) is the proxy reconstruction of ΔGMST by [9] projected to 5˚ × 5˚ resolution. The blue line in (d) is the DA-based reconstruction of ΔGMST by [25].

When we consider the global mean temperatures, the DA solution in Exp_GLAC1D closely resembles the prior states (Fig 6d), but the warming trend reaches its maximum at the beginning of YD, with a reduction of approximately 0.2 degrees compared to the pre-DA state. A robust cooling trend is also observed in YD. These abrupt changes are consistent with AMOC variations. For Exp_PaleoMist (Fig 7d), the DA trajectory closely follows the global temperature of the prior state, implying that the paleo-observations have a minor effect on ΔGMSTs during the last deglaciation when using the PaleoMist reconstruction. We mention that the DA, prior and free trajectories exhibit two sudden upward shifts during the early Holocene in Exp_PaleoMist. This unrealistic feature can be attributed to the low temporal resolution of the PaleoMist reconstruction, which is 2500 years.

When comparing our DA solutions with [25] reconstruction (Osman-DA; Figs 6d and 7d), our DA ΔGMST trajectories are generally warmer then Osman-DA during the deglaciation. Specifically, when focusing on BA and YD, the timing and magnitude of these events in Exp_GLAC1D differ notably from the Osman-DA, but the DA pattern in Exp_PaleoMist is similar to that in the Osman-DA. However, the maximum warming in BA for Exp_PaleoMist occurs approximately 100 years earlier than in the Osman-DA. It is clear that discrepancies between our results and Osman-DA are due to utilising different observation datasets, background states, and methods.

## Surface temperature fields after DA

The DA has a significant effect on the pattern of the deglacial temperature evolution. Fig 8 shows that the effect of DA is more pronounced at mid- and high-latitudes (from 3˚C to more than 5˚C) but small at low latitudes (less than 2˚C). Comparing Exp_GLAC1D with Exp_PaleoMist, we see that the DA system has a different effect in some regions due to the use of different ice sheet reconstructions. For example, in Exp_GLAC1D during YD (Fig 8c), Antarctica transitions to a colder state after DA, while it becomes warmer in Exp_PaleoMist (Fig 8f). During BA, the discrepancies between the DA experiments are remarkable. In contrast to Exp_GLAC1D (Fig 8b), Exp_PaleoMist (Fig 8e) shows strong cooling over the North Atlantic and warming over Antarctica and the Southern Ocean. Moreover, after DA, both experiments improve the Atlantic-Pacific seesaw phenomenon (e.g. [60–62]), characterized by the opposing temperature anomalies observed in the North Atlantic and North Pacific Ocean regions, except for the BA in Exp_GLAC1D.

The BA is marked by a pronounced warming in the Northern Hemisphere, particularly over Greenland ([63]). Comparing the ΔGMST anomaly of BA against that of LGM (Figs 9a, 9d, 10a and 10d), the warming over Greenland and the North Atlantic increases by almost 5˚ with DA in both experiments. However, there are some cooling shifts over the North Pacific (Figs 9d and 10d), which is more evident in Exp_GLAC1D.

In contrast to the BA, the average temperature in the Northern Hemisphere decreased by several degrees during the YD, resulting in a return to near-glacial conditions. In the Southern Hemisphere, the YD temperature did not vary significantly and was comparable to or even slightly warmer than BA ([64, 65]). These properties of YD are enhanced by the implementation of DA. Figs 9f and 10f show that the Northern Hemisphere climate in YD changes to a colder state after DA than in BA. Nevertheless, this change is more pronounced in Exp_PaleoMist. Without DA, the North Atlantic temperature drop between BA and YD could not be simulated.

## Discussion

To overcome the computational challenges associated with long-term DA experiments in paleoclimate studies, we have coupled CLIMBER-X with PDAF. Our DA experiments

## Exp_GLAC1D vs Exp_PaleoMist

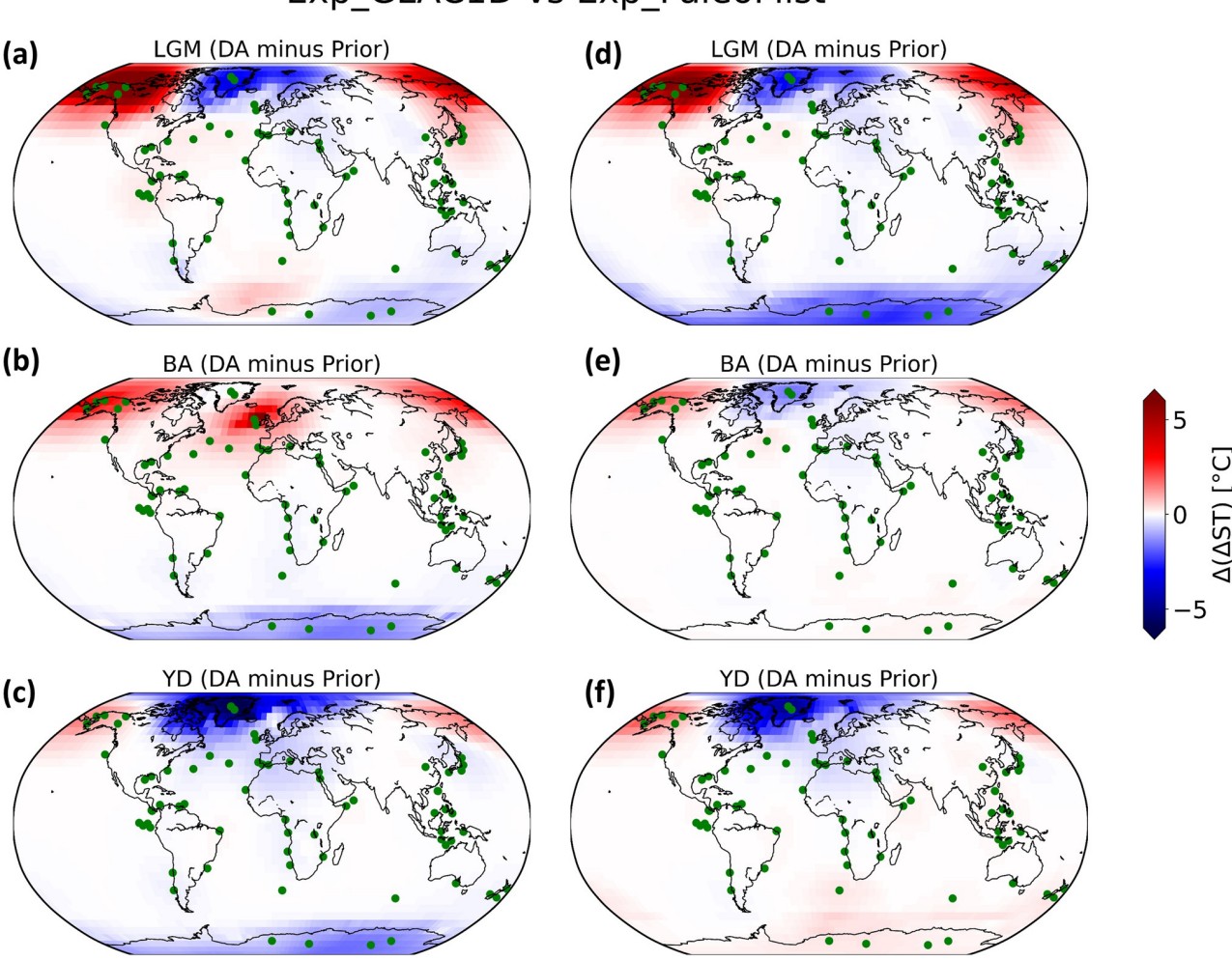

**Fig 8. Effect of DA on ST fields in Exp_GLAC1D and Exp_PaleoMist.** ΔST anomaly (DA minus Prior) in LGM, BA, and YD. (a), (b), and (c) show anomaly for Exp_GLAC1D and (d), (e), and (f) for Exp_PaleoMist. The green dots indicate the observation locations.

covering a span of 16500 years can be completed in approximately 40 hours. PDAF efficiently computes the DA update at each DA cycle in less than one second.

The choice of localization radius is an important factor in our DA system, and we determined the optimal value through trial and error. The optimal localization radius depends on various factors, including ensemble size, the spatial distribution of observations, and characteristics of observation errors ([66]). [67] has shown that the optimal localization radius remains unaffected by properties of a quasigeostrophic model ([68]), such as resolution, as long as the model accurately represents the underlying dynamical processes. As discussed in Experimental Design section, we selected the 5000 km radius to ensure the optimal impact of observations while avoiding unrealistic influences from distant observations on individual grid points. This radius mainly preserves the effect of Greenland's observations, which are crucial for reconstructing the North Atlantic. The choice of localization radii for DA experiments involving CLIMBER-X and PDAF may vary depending on the specifics of the observation network.

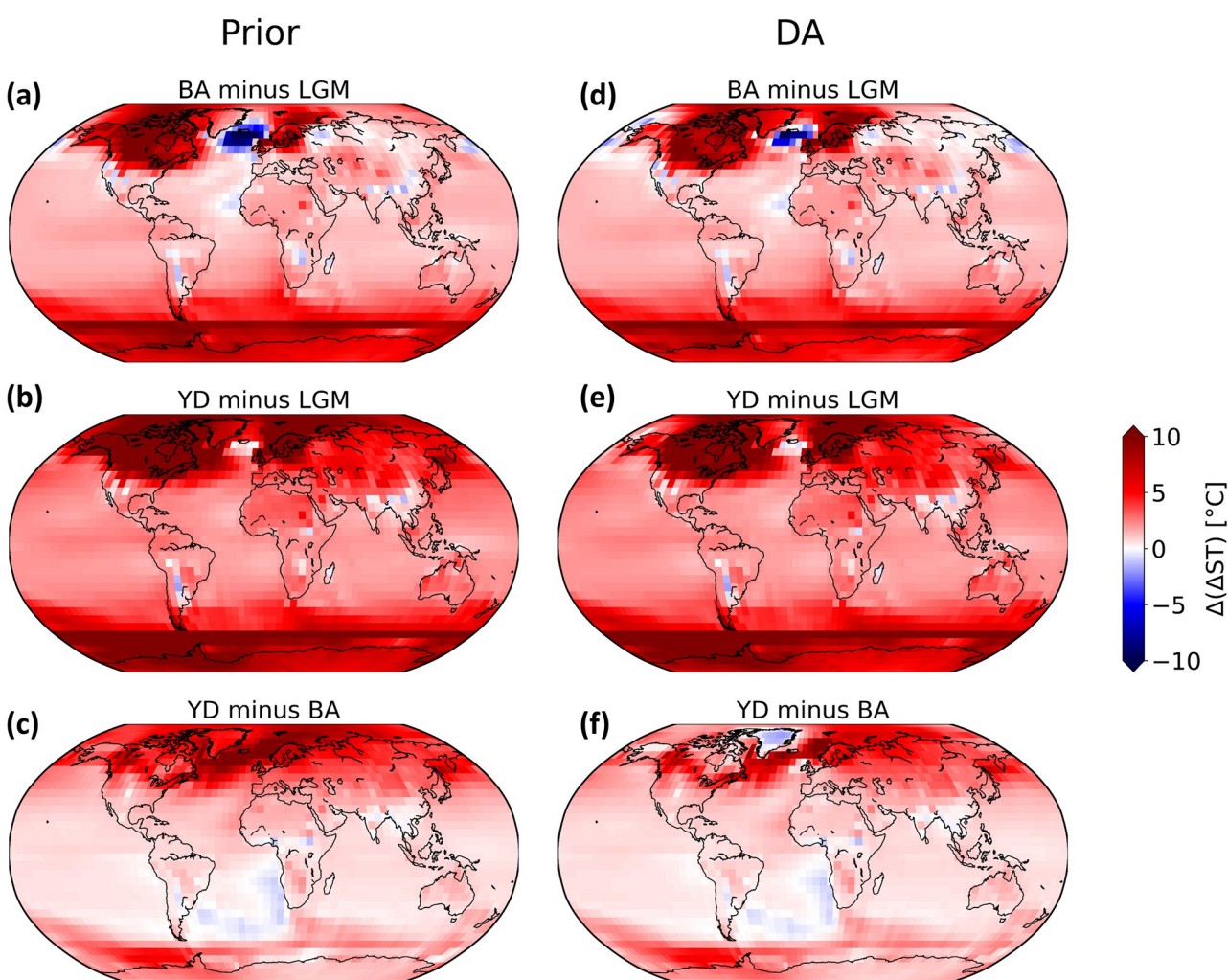

**Fig 9. Comparison of ST anomalies in Exp_GLAC1D before and after DA.** ΔST anomalies field for different time intervals in Exp_GLAC1D for before DA's implementation (a), (b), and (c) and after DA (d), (e), and (f).

A climate system consists of slowly varying components (e.g., ocean, cryosphere, land vegetation) and fast-varying components, primarily the atmosphere. The deterministic models, including CLIMBER-X, typically operate in a deterministic framework, where the averaged climate states are simulated based on deterministic mathematical equations, fixed initial conditions, external forcings, and parameterizations of fast-varying components ([69]). In contrast, stochastic climate models acknowledge the importance of considering the rapid fluctuations and introduce randomness to capture uncertainties and natural variability in the climate system ([70]). [71] emphasizes the significance of incorporating high-frequency elements in numerical models, as slow climate changes are defined as the response to ongoing random excitation by fast-varying component perturbations. By perturbing the near atmospheric surface temperature as a rapid-varying variable in the CLIMBER-X model, we have transformed it into a stochastic model, allowing for the representation of natural variability and uncertainties. This approach not only leverages the computational efficiency of CLIMBER-X but also provides more realistic background states for our DA system and avoids the ensemble collapses to a single state.

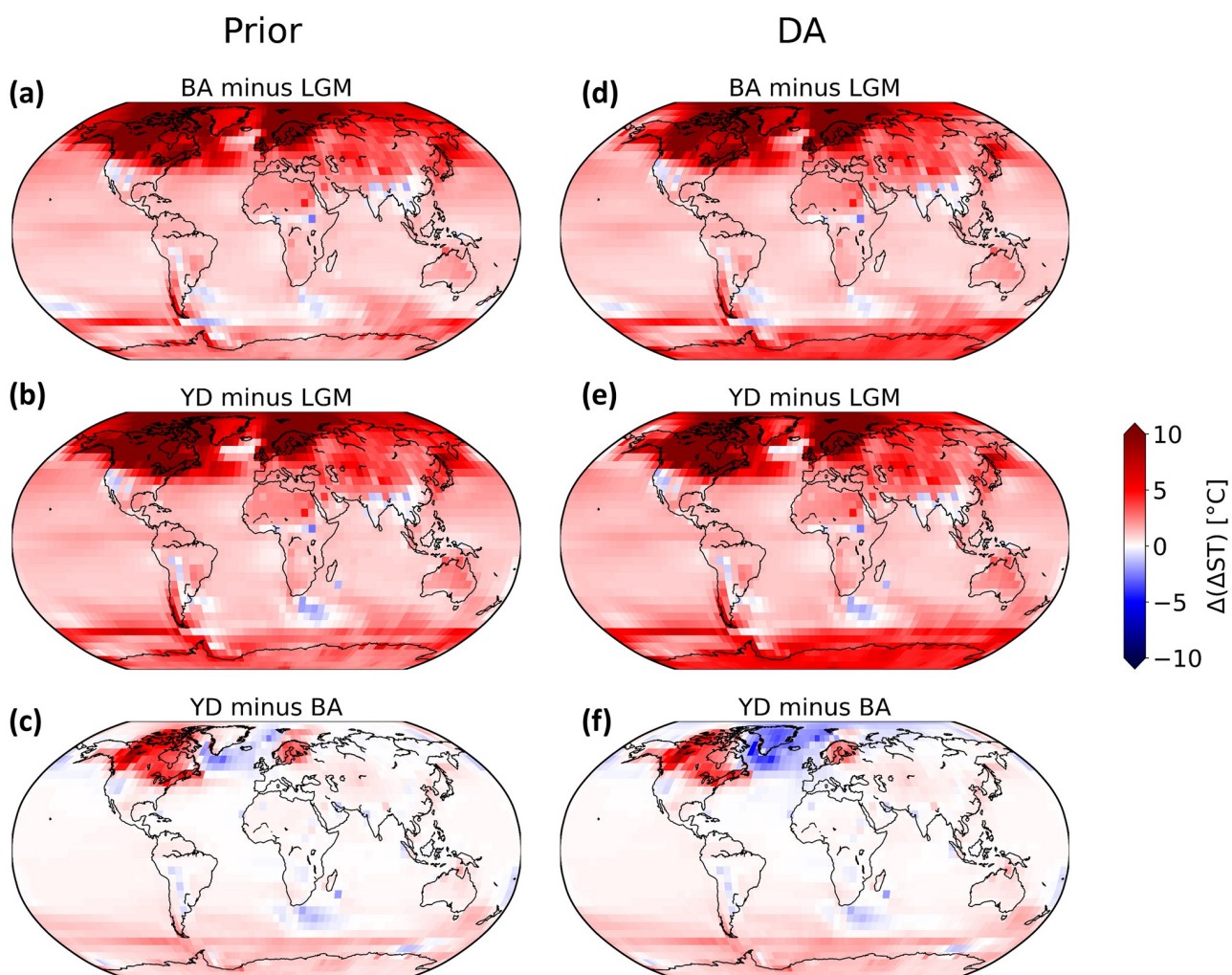

**Fig 10. Comparison of ST anomalies in Exp_PaleoMist before and after DA.** ΔST anomalies field for different time intervals in Exp_Paleomist for before DA's implementation (a), (b), and (c) and after DA (d), (e), and (f).

Climate models, including CLIMBER-X, are sensitive to freshwater fluxes ([72–74]), and the AMOC depends on freshwater forcing at locations where deep water forms ([75]). Accordingly, changes in AMOC are often considered to have caused abrupt climate changes during the last deglaciation (e.g. [4, 59, 76, 77]). GLAC-1D contains a significant ice volume loss during BA ([42]), associated with a pronounced meltwater pulse 1A (MWP-1A), whereas Paleo-Mist ([43]) does not contain a pronounced loss of deglacial meltwater during the BA. This is the main reason for the discrepancies between Exp_GLAC1D and Exp_PaleoMist. Moreover, ice sheet heights influence atmospheric and oceanic circulation (e.g. [78, 79]), which may contribute to the disparities between simulations using GLAC-1D and PaleoMist. Due to variations in methodologies, PaleoMist estimates the ice volume and ice sheet heights differently from GLAC-1D. Mentioning briefly, PaleoMist calculates ice sheets using the program ICE-SHEET, which assumes that the ice sheet is in equilibrium ([80]). In contrast, GLAC-1D calculates Antarctica and North American ice sheets based on an ensemble average of several thousand ice sheet model simulations that fit constraints such as Holocene sea level changes and present-day uplift rates ([42, 81]).

The minor effect of DA on GMST trajectories could be attributed to the limited number of observations and the long period between DA cycles. The time gap between successive observations, 100 years, may exceed the model's predictability. Consequently, the prior ensemble loses the updated initial condition information entirely. This point is also mentioned by [52, 82] while comparing online and offline approaches. Therefore, in our experiments, GMST trajectories are mainly driven by external forcing, including ice sheets, GHGs, and orbital forcings. Increasing the frequency of model updates by employing high-temporal-resolution, well-distributed datasets covering more grid points can potentially improve the efficiency and impact of DA on GMST trajectories. However, the ΔGMST trends for prior and DA in Exp_PaleoMist are similar to the Shakun et al. reconstruction and Osman-DA during BA and YD. This indicates that the choice of ice-sheet reconstruction resulting in different prior states is consequential in our DA system.

[25] employ an offline ensemble square root Kalman filter approach ([23]) and assimilate the different types of geochemical proxies for sea surface temperature ([23]) directly using Bayesian proxy forward models (e.g. [83]). They also draw the prior states from the separate time slice simulations conducted by the isotope-enabled Community Earth System Model (iCESM; [84]). These methodological disparities between their DA setup and ours explain the differences between the results. Nevertheless, Exp_PaleoMist is roughly consistent with Osman-DA regarding the magnitude and speed of the warming-cooling-warming tendency over BA and YD. This suggests that despite the differences in methodology and background states, there is some agreement regarding the general pattern of GMST changes during those time intervals.

In contrast to the effect of DA on GMST trajectories, DA has changed the spatial pattern of the ST fields significantly. This implies that the impact of DA is more prominent at regional scales rather than at the global mean scale. The point that the climate sensitivity of CLIMBER-X is 3.3 K, which drops in the middle range for the Earth system models ([29]), suggests that the model is responsive to $CO_2$ forcing and insolation. As a result, the main drivers in the low latitudes, such as GHG concentrations and insolation, dominate in shaping the temperature patterns in those regions. Consequently, the effect of DA is less pronounced in the low latitudes where the model already well-captured these primary drivers.

However, in high latitudes, particularly over the North Atlantic and Greenland, where the ice sheet heights, FW pattern and AMOC play significant roles in atmospheric and oceanic circulations and temperature patterns, DA enhances the characteristics of BA and YD. This denotes that the assimilation of observational data has helped skillfully to have more realistic temperature patterns and climate variability during these specific periods.

Additionally, it is worth mentioning that DA has enhanced the Atlantic-Pacific seesaw pattern, mainly for Exp_PaleoMist, indicating an improved representation of the coupled atmosphere-ocean dynamics in these regions. Several studies have reported this heterogeneous phenomenon in both model and data sets during glacial and deglacial periods (e.g. [61, 85, 86]). Different factors, including oceanic and atmospheric circulation patterns, heat transport, and interactions between the oceans and atmosphere, contribute to the evolution and endurance of this seesaw pattern (e.g. [62, 87–89]). Therefore, the assimilation of observational data has contributed to a more realistic simulation of the interhemispheric temperature gradient and associated oceanic circulation patterns.

## Conclusions

For the last deglaciation, there are various challenges in reconstructing past climates, including uncertainties in proxy data, limited spatial and temporal coverage of observations, and

complex interactions between different climate system components. Data assimilation (DA) helps to address these challenges by integrating available observations with model simulations, taking into account their respective uncertainties and biases.

We introduce a fast and efficient method for conducting DA using an EMIC, CLIMBER-X ([29]). Since CLIMBER-X has no internal noise in the system, we applied a stochastic version of this model. In addition, we use two different ice sheet reconstructions, GLAC-1D ([42]) and PaleoMist ([43]), to investigate the effects of different model backgrounds on the climate evolution during the last deglaciation. We summarize the conclusions of our work in the following main points:

- The choice of ice sheet reconstruction significantly impacts model simulations, affecting ocean freshwater forcings and AMOC, leading to different circulation and temperature patterns during the deglaciation. The free simulation with the PaleoMist ice sheet reconstruction provides a more consistent trend than that with GLAC-1D, especially during the Bølling-Allerød—Younger Dryas sequence. Accordingly, the ice sheet reconstructions lead to different effects of the DA.

- While DA has a minor effect on global mean surface temperature trajectories, it has significantly influenced the surface temperature fields, suggesting that the impact of DA is more prominent at regional scales rather than at the global mean scale. Our DA system improves the ST spatial heterogeneity (Atlantic-Pacific seesaw), representing the climate patterns for YD and BA, especially for the PaleoMist experiment.

- The effect of DA is more pronounced at high latitudes than at mid and low latitudes, potentially indicating disparities or inadequate representation of physical processes within the model. Considering its climate sensitivity, CLIMBER-X performs relatively more accurately in low latitudes, where the main driver is $CO_2$ forcing, compared to high latitudes.

- Our online DA approach allows us to study the performance of CLIMBER-X, including AMOC, salinity, freshwater, and many other climate parameters. Nevertheless, the effect of the online approach is not notable on the climate variables primarily due to the long time gaps between DA cycles and due to the fact that temperature is damped out faster than other variables like salinity ([90]).

Our method is a step towards a true paleoclimate data assimilation. We note that the methodology presented in this work is not restricted to our specific application of deglacial climate. CLIMBER-X and PDAF or their variations can also be applied to other data sets and scientific questions for long-term variations. Thus, developing this methodology is a scientific contribution in its own right. As a logical next step, we will insert subsurface data and salinity in our assimilation in order to evaluate the importance of subsurface temperatures as potential predictors for abrupt changes in deglacial AMOC ([4, 12, 91]).

## Supporting information

**S1 Fig. GLAC-1D reconstruction for the last deglaciation.**
(TIF)

**S2 Fig. PaleoMist reconstruction for the last deglaciation.**
(TIF)

**S3 Fig. GHG forcing.**
(TIF)

## Acknowledgments

The authors wish to thank the AWI computing center for their help. Thanks go to Hugues Goosse for giving advice on our data assimilation approach.

## Author Contributions

**Conceptualization:** Ahmadreza Masoum, Lars Nerger, Gerrit Lohmann.

**Investigation:** Ahmadreza Masoum.

**Methodology:** Ahmadreza Masoum, Lars Nerger, Gerrit Lohmann.

**Software:** Lars Nerger, Matteo Willeit, Andrey Ganopolski.

**Supervision:** Lars Nerger, Gerrit Lohmann.

**Visualization:** Ahmadreza Masoum.

**Writing – original draft:** Ahmadreza Masoum.

**Writing – review & editing:** Ahmadreza Masoum, Lars Nerger, Matteo Willeit, Andrey Gano-polski, Gerrit Lohmann.

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
