## [Decision Letter · Decision Letter 0]

2 Nov 2023

PONE-D-23-19461Paleoclimate Data Assimilation with CLIMBER-X: An ensemble Kalman filter for the Last DeglaciationPLOS ONE

Dear Dr. Masoum,

Thank you for submitting your manuscript to PLOS ONE. After careful consideration, we feel that it has merit but does not fully meet PLOS ONE’s publication criteria as it currently stands. Therefore, we invite you to submit a revised version of the manuscript that addresses the points raised during the review process.

We look forward to receiving your revised manuscript.

Kind regards,

Yougui Song

Academic Editor

PLOS ONE

Journal Requirements:

3. We note that Figures 2,8,9 & 10 in your submission contain [map/satellite] images which may be copyrighted. All PLOS content is published under the Creative Commons Attribution License (CC BY 4.0), which means that the manuscript, images, and Supporting Information files will be freely available online, and any third party is permitted to access, download, copy, distribute, and use these materials in any way, even commercially, with proper attribution. For these reasons, we cannot publish previously copyrighted maps or satellite images created using proprietary data, such as Google software (Google Maps, Street View, and Earth). For more information, see our copyright guidelines: http://journals.plos.org/plosone/s/licenses-and-copyright.

 a. You may seek permission from the original copyright holder of Figures 2,8,9 & 10 to publish the content specifically under the CC BY 4.0 license. 

Additional Editor Comments (if provided):

I hope this message finds you well. I wanted to extend my sincerest apologies for the delay in the peer review process for your manuscript. Despite our efforts, we have encountered challenges in securing potential reviewers including those you had suggested, and obtaining comments from reviewers who initially agreed to evaluate your work. By now, we only obtained one feedback, comments. Considering timeliness, we have decided to forward the comments to you. Although this is not the ideal scenario, we believe that the provided feedback can still be valuable to you for making necessary revisions to your manuscript.

Reviewers' comments:

Reviewer's Responses to Questions

**Comments to the Author**

1. Is the manuscript technically sound, and do the data support the conclusions?

Reviewer #1: Yes

2. Has the statistical analysis been performed appropriately and rigorously? 

Reviewer #1: Yes

3. Have the authors made all data underlying the findings in their manuscript fully available?

Reviewer #1: Yes

4. Is the manuscript presented in an intelligible fashion and written in standard English?

Reviewer #1: Yes

5. Review Comments to the Author

Reviewer #1: Review of Paleoclimate Data Assimilation with CLIMBER-X: An ensemble Kalman filter for the Last Deglaciation

This paper, using the climate model CLIMBER-X, presents an method for online method to assimilate the temporal evolution of surface temperatures for the last deglaciation. This paper is interesting for paleoclimate reconstructions and well written. However, I have two major comments about this manuscript and it may be suitable to publish in this journal after major revions.

1, As we know, large amounts of paleoclimate DA methods are most based on the offline DA. That is because the observational records only provide the mean values (e.g., annual mean surface temperature) instead of instantaneous values. I think the authors should address how initial conditions change associated with annual mean surface temperature in the online DA. Additionally, it has 100 year period in the online cycle, I am not sure that the initial conditions of surface temperature is important for the surface temperature after 100 years. I think it is mainly dependent on external forcing. I suggest the authors should claim these points to increase the readability of the article.

2, In the manuscript, the localization radius is set to 5000 km. It seems too small. In the previous studies, they used large localization radius (such as 25000km and even infinite) to assimilate the global surface temperature. Therefore, I suggest the authors should evaluate the results according to observations and choose the best radius (shown in King et al., 2021, JC).

King et al., A Data Assimilation Approach to Last Millennium Temperature Field Reconstruction Using a Limited High-Sensitivity Proxy Network.

6. PLOS authors have the option to publish the peer review history of their article (what does this mean?). If published, this will include your full peer review and any attached files.

Reviewer #1: No

---

## [Author Response · Author response to Decision Letter 0]

18 Dec 2023

Authors’ Response to the Editor:

Thank you for your message and sincere apologies for the challenges faced in the peer review process for my manuscript. We understand the difficulties in finding reviewers and appreciate your handling of the review process and your decision to share the obtained feedback despite the limitations. According to the reviewer's comments, we have checked our manuscript and addressed them by adding content to the Experimental Design and Discussion sections of the manuscript.

Authors’ Response to Reviewer 1:

Thank you for your feedback and comments. We have carefully addressed all the issues item by item as follows.

Response to comment 1:

We think this comment consists of two points: the reason for the choice of the online approach and the influence of updating initial conditions in our DA method. We believe that the main reason for the domination of the offline approach in paleoclimate DA is the high computational costs of the online DA. CLIMBER-X as an earth system model of intermediate complexity (EMIC), provides us the opportunity to apply online DA. Moreover, the online DA approach allows us to study the performance of CLIMBER-X, including AMOC, salinity, freshwater, and many other climate parameters. Therefore, we decided to apply a more complex method, online DA, because we are able to do that, and it has the potential to get better results than offline DA. To clarify this point, We added more explanations to the manuscript, lines 164-177.

 In reply to the second point, we agree that the time gap between successive observations, 100 years, may exceed the model's predictability. Thus, over 100 years we do not expect that the initial state (previous analysis) has an influence. We have added more explanations to the Discussion section, lines 375-380, and clearly mentioned that the surface temperature is mainly driven by external forcings in our experiments. However, our DA method can use high temporal resolution or artificial observations for different applications. For example, one could apply a 10-year temporal resolution just by interpolating the observation values. Then, the effect of temperature could probably persist.

Response to comment 2 :

The localization radius is always case-dependent, and there are different ways to determine an optimal radius. Depending on different factors, such as the prior states, DA approach, observation, and the targets of the articles, the previous studies have defined their own criteria and selected different localization radii. For example, the offline DA reconstructions such as Erb et al. (2022), King et al. (2021), Osman et al. (2021), and Tierney et al. (2021) who use outputs of different coupled general circulation models for generating ensemble priors, select a relatively large radius from 12000 km to 25000 km. However, Okazaki et al. (2021) employ an EMIC for their online DA, using a 2000, 5000, and 8000 km radius.

 Our main criterion is the effect of DA on the surface temperature field. During our trials, we found when we increased the radius, the North Atlantic area would be dominated by the observations located in the northwest of North America, and the effect of Greenland's observation would be lost after DA. Therefore, we decided to select a localization radius of 5000 km. Clearly, one could choose another localization radius with different observation networks. We have added further explanations in the Experimental Design, lines 183-190, and Discussion sections, lines 337-341, to address this comment and improve the readability of the paper.

References :

 Tierney, J. E., Zhu, J., King, J., Malevich, S. B., Hakim, G. J., and Poulsen, C. J. (2020). Glacial cooling and climate sensitivity revisited. Nature, 584(7822):569–573.

 Erb, M. P., McKay, N. P., Steiger, N., Dee, S., Hancock, C., Ivanovic, R. F., Gregoire, L. J., and Valdes, P. (2022). Reconstructing holocene temperatures in time and space using paleoclimate data assimilation. Climate of the Past, 18(12):2599–2629.

 King, J. M., Anchukaitis, K. J., Tierney, J. E., Hakim, G. J., Emile-Geay, J., Zhu, F., and Wilson, R. (2021). A data assimilation approach to last millennium temperature field reconstruction using a limited high-sensitivity proxy network. Journal of Climate, 34(17):7091–7111.

 Okazaki, A., Miyoshi, T., Yoshimura, K., Greybush, S. J., and Zhang, F. (2021). Revisiting online and offline data assimilation comparison for paleoclimate reconstruction: an idealized osse study. Journal of Geophysical Research: Atmospheres, 126(16):e2020JD034214.

 Osman, M. B., Tierney, J. E., Zhu, J., Tardif, R., Hakim, G. J., King, J., and Poulsen, C. J. (2021). Globally resolved surface temperatures since the last glacial maximum. Nature, 599(7884):239–244.

---

## [Decision Letter · Decision Letter 1]

23 Feb 2024

Paleoclimate Data Assimilation with CLIMBER-X: An ensemble Kalman filter for the Last Deglaciation

PONE-D-23-19461R1

Dear Dr. Masoum,

We’re pleased to inform you that your manuscript has been judged scientifically suitable for publication and will be formally accepted for publication once it meets all outstanding technical requirements.

Kind regards,

Yougui Song

Academic Editor

PLOS ONE

Additional Editor Comments (optional):

Reviewers' comments:

Reviewer's Responses to Questions

**Comments to the Author**

1. If the authors have adequately addressed your comments raised in a previous round of review and you feel that this manuscript is now acceptable for publication, you may indicate that here to bypass the “Comments to the Author” section, enter your conflict of interest statement in the “Confidential to Editor” section, and submit your "Accept" recommendation.

Reviewer #1: All comments have been addressed

2. Is the manuscript technically sound, and do the data support the conclusions?

Reviewer #1: Yes

3. Has the statistical analysis been performed appropriately and rigorously? 

Reviewer #1: Yes

4. Have the authors made all data underlying the findings in their manuscript fully available?

Reviewer #1: Yes

5. Is the manuscript presented in an intelligible fashion and written in standard English?

Reviewer #1: Yes

6. Review Comments to the Author

Reviewer #1: Review of "Paleoclimate Data Assimilation with CLIMBER-X: An ensemble Kalman filter for the Last Deglaciation"

Thank the authors for their responses. It is okay for me.

7. PLOS authors have the option to publish the peer review history of their article (what does this mean?). If published, this will include your full peer review and any attached files.

Reviewer #1: **Yes: **Xiaoning Xie

---

## [Editor Report · Acceptance letter]

25 Mar 2024

PONE-D-23-19461R1 

PLOS ONE

Dear Dr. Masoum, 

I'm pleased to inform you that your manuscript has been deemed suitable for publication in PLOS ONE. Congratulations! Your manuscript is now being handed over to our production team.

Kind regards, 

on behalf of

Dr. Yougui Song 

Academic Editor

PLOS ONE